# Diverse Communities of Endemic and Cosmopolitan Lineages at Local Sites in the *Lecanora polytropa* Aggregate (Ascomycota)

John Woodhouse [1], Sergio Pérez-Ortega [2], Claude Roux [3], Michel Bertrand [4] and Steven D. Leavitt [5,*]

1 Department of Biology, Brigham Young University, Provo, UT 84602, USA
2 Department of Mycology, Real Jardín Botánico (CSIC), Claudio Moyano 1, 28014 Madrid, Spain; sperezortega@rjb.csic.es
3 390 Chemin des Vignes Vieilles, 84120 Mirabeau, France; claude.roux21@wanadoo.fr
4 La Grande Bastid, 84750 Viens, France; bertrand.mic@wanadoo.fr
5 Department of Biology & M.L. Bean Life Science Museum, Brigham Young University, Provo, UT 84602, USA
* Correspondence: steve_leavitt@byu.edu; Tel.: +1-801-422-4879

**Abstract:** Recent work has suggested exceptional species-level diversity in the lichen-forming *Lecanora polytropa* complex (*Lecanoraceae*, *Ascomycota*). However, biogeographic patterns and the spatial structuring of this diversity remains poorly known. To investigate diversity across multiple spatial scales, we sampled members of this species complex from two distinct regions—the Pacific Coast Ranges in southern Alaska, USA, and montane habitats in Spain. We also included sequence data from several species within this complex that were recently described from populations in France. Using the standard DNA barcoding marker and a sequence-based species delimitation approach (ASAP), we inferred a total of 123 candidate species (SHs) within the *Lecanora polytropa* complex, 32 of which were sampled for the first time here. Of 123 SHs, 21 had documented intercontinental distributions, while the vast majority were found at much smaller spatial scales. From our samples collected from Alaska, USA, and Spain, representing 36 SHs, we found high genetic diversity occurring within each sampled site, but limited overlap among all sites. Mountain ranges in both regions had high proportions of endemic lineages, with the highest diversity and endemism occurring in mountain ranges in Spain. Our sequence data generally support the recent taxonomic proposals, and an integrative taxonomy may help partly resolve the taxonomic conundrums within this hyper-diverse lineage.

**Keywords:** ASAP; glacial refugia; hyper-diverse; integrative taxonomy; *Lecanoraceae*; species delimitation

## 1. Introduction

Lichens, which represent a complex symbiotic relationship between fungi, photosynthetic partners (algae or cyanobacteria), and other microorganisms [1], offer a unique system by which to investigate the origin of diversity and biogeographic patterns [2–4]. Understanding their diversity and geographic distributions can aid in deciphering the mechanisms driving symbiosis, speciation, and adaptation. This is particularly relevant in the context of environmental changes, where lichens act as sensitive indicators [5–7]. For some lichens, diversity of the lichen symbionts is poorly understood, masked by limited diagnostic features [8–10], convergent morphologies [11], and the complexities of symbiotic interactions influencing lichen phenotypes [12]. The occurrence of taxonomically problematic species is likely non-random in relationship to taxon and biome and has implications for evolutionary theory, biogeography and conservation [13,14].

Recent work has revealed exceptional diversity in a common, widespread lineage of lichen-forming fungi—the *Lecanora polytropa* (Ehrh.) Rabenh. complex, which also includes the *L. intricata* (Ach.) Ach. group (*Ascomycota*, *Lecanoraceae*) [15]. Members of the *L. polytropa* group comprise non-placodiod *Lecanora* species with a yellow-green- or pale yellow-brown-colored thallus and thalline rim and produce usnic acid (or isousnic acid) [16]. This clade comprises ca. 30 different species, including a number of recently described

taxa [16]. In contrast, Zhang et al. [15] have reported data that suggest the potential of nearly one hundred species-level lineages within this group. Despite its common occurrence and general recognition by lichen researchers [17], the *L. polytropa* complex has received relatively little systematic consideration, and, to date, no comprehensive taxonomic treatment is available for this group. However, studies continue to reveal additional insight into the evolutionary relationships and range of diversity in some members of this clade [18–21].

Although recent sampling efforts have revealed unexpected genetic diversity across populations worldwide [15], it is uncertain to what extent the total diversity of species-level lineages in the *L. polytropa* complex is represented. Furthermore, patterns of genetic diversity at various spatial scales are not known. Some species-level lineages in the *L. polytropa* complex have confirmed intercontinental distributions, while others appear to be restricted to local scales [15]. To investigate whether current sampling and representative sequence data comprehensively captures diversity within the *L. polytropa* complex, here we generated sequence data from members of this group from two regions that have not been included in earlier DNA-based studies—the Pacific Coast Ranges in southern Alaska, USA, and Mediterranean and Atlantic montane habitats in Spain (Figure 1). We also included representatives from a number of recently described species [16]. Based on this new sampling, we (i) investigated species-level diversity of members of the *L. polytropa* complex at local scales, (ii) compared diversity among sampled sites to gain insight into the proportion of endemic lineages relative to cosmopolitan lineages, and (iii) tested whether some recently described species are supported by genetic sequence data. Our results highlight that species-level diversity in the *L. polytropa* complex remains incompletely sampled, and that unsampled regions harbor high levels of genetic diversity. By including representatives of recently described species in our molecular sequence dataset, we show that integrative taxonomic approaches may help establish a stable, robust taxonomy for the *L. polytropa* complex.

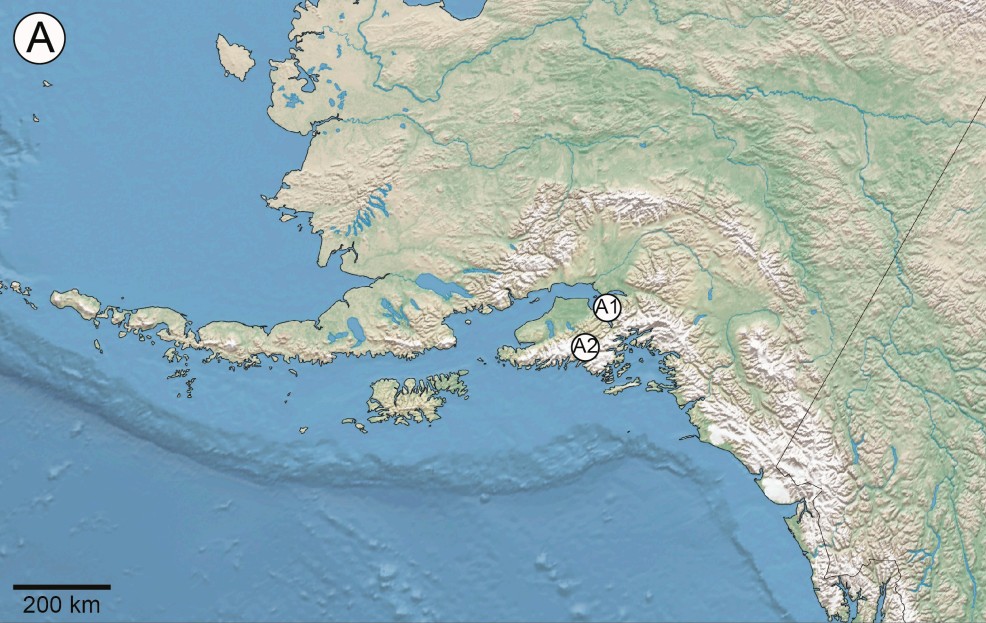

**Figure 1.** *Cont.*

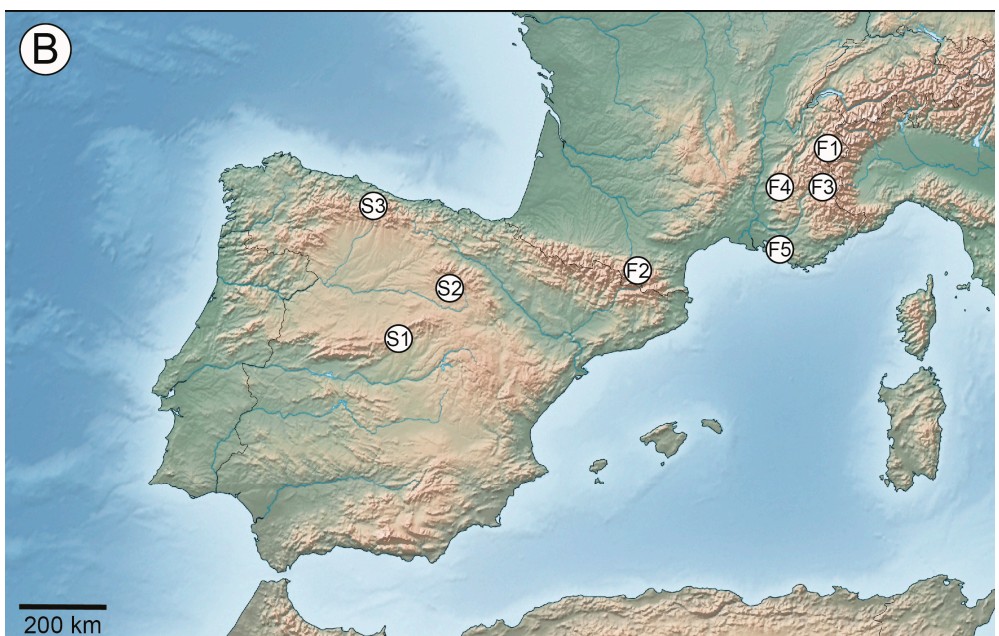

**Figure 1.** (**A**) Sampling areas in the Pacific Coast Range of Alaska, USA—'A1', Chugach Mountains and 'A2' Kenai Mountains; (**B**) sampling areas in Mediterranean and Atlantic montane habitats in Spain—'S1' Sierra de Guadarrama in the center of the Central System, 'S2' in the Parque Natural de la Laguna Negra and the Circos Glaciares de Urbión in the Iberian System, and 'S3' in Ponga Biosphere in the Cantabrian Mountains. Specimens of taxonomic interest were sampled in France from the following locations: 'F1' French Alps, in the National Park of the Vanoise; 'F2' French Pyrenees; 'F3' French Prealps, in the Queyras Regional Park; 'F4' Provence Prealps, on Mont Ventoux; and 'F5' Provence, on the rocky, siliceous Mediterranean coast.

## 2. Materials and Methods

### 2.1. Taxon Sampling

For this study, we sampled members of the *Lecanora polytropa* complex *sensu* [15] from two distinct regions—the Pacific Coast Ranges in southern Alaska, USA, and Mediterranean and Atlantic montane habitats in Spain (Figure 1). In Alaska, specimens were collected from two sites—"A1" in the Chugach Mountains in the northwestern part of the Gulf of Alaska near Anchorage, and "A2" in the Kenai Mountains (Dena'ina: Yaghanen Dghili) near Seward, separated by approximately 115 km (Figure 1A). At both collection locations, specimens were opportunistically collected, in an attempt to represent the range of phenotypic variation observed within the field. In Spain, specimens were collected from three mountain ranges—"S1" in the Sierra de Guadarrama in the center of the Central System, "S2" in the Parque Natural de la Laguna Negra y los Circos Glaciares de Urbión in the Iberian System, and "S3" in the Ponga Biosphere in the Cantabrian Mountains. The three sampling locations were separated from each other by 160–270 km (Figure 1B). Similarly, specimens were opportunistically collected at the three locations, in an attempt to represent the range of phenotypic variation observed within the field. For all new collections from Alaska, USA and Spain, provisional determinations were made following [16], with specimens representing *L. alpigena* (Ach.) Cl. Roux, *L. dispersoareolata* (Schaer.) Lamy, *L intricata* (Ach.) Ach., *L. polytropa* (Hoffm.) Rabenh., and *L. silvae-nigrae* V. Wirth. Recent detailed taxonomic work on members of the *L polytropa* group has resulted in a number of new taxonomic entities [16]. Here, we also included an additional eleven specimens of taxonomic interest from southern France based on that work, as follows: *L. crassiuscula* Cl. Roux, Poumarat et M. Bertrand (*n* = 1), *L. dispersoareolata* (*n* = 2), *L. dispersosquamulata* ad int. (*n* = 2), *L. hyperpolytropa* Cl. Roux et M. Bertrand (*n* = 1), *L. polytropa* (*n* = 1), *L. polytropopsis* Cl. Roux, M. Bertrand et Poumarat (*n* = 1), *L. stenotropa* Nyl. (*n* = 2), and *L. vocontia* (Clauzade et Cl. Roux) Cl. Roux et M. Bertrand (*n* = 1).

To investigate broad-scale distribution patterns, sequence data from newly collected specimens were combined with sequences from 340 additional *L. polytropa sensu lato* specimens collected from worldwide populations [15]. The final dataset comprised a total of 510 sequences from specimens within the *L. polytropa* complex (Supplementary File S1).

### 2.2. DNA Extraction, Sequencing and Alignment

Total genomic DNA were extracted from specimens collected for this study using the Wizard Genomic DNA Purification Kit (Promega, Madison, WI, USA). We attempted to generate sequence data from the internal transcribed spacer region (ITS, ITS1, 5.8S, ITS2) [22] using primers ITS1f with ITS4 [23]. The temperature profile for the polymerase chain reaction (PCR) amplifications follows [24]. PCR amplifications were performed using Cytiva PuReTaq Ready-To-Go™ PCR Beads (ThermoFisher Scientific, Waltham, MA, USA), and products were visualized on 1% agarose gel. Successful amplifications were cleaned using ExoSAP-IT Express (ThermoFisher Scientific), following the manufacturer's recommendations. Sequencing reactions for both complementary strands were performed using BigDye 3.1 (Applied Biosystems, Foster City, CA, USA) and run on an ABI 3730 automated sequencer (Applied Biosystems) at the DNA Sequencing Center at Brigham Young University, Provo, UT, USA. ITS sequences generated for this study were combined with those from [15] and aligned using the program MAFFT v7 [25,26]. We implemented the G-INS-i alignment algorithm and '1PAM/K = 2' scoring matrix, with an offset value of 0.1, the 'unalignlevel' = 0.2, and the remaining parameters were set to default values.

### 2.3. Candidate Species Delimitation Using the Standard Fungal DNA Barcode

Initial candidate species partitions, or species hypotheses (SHs), for the *L. polytropa* group were inferred using the Assemble Species by Automatic Partitioning (ASAP) method [27], based on the multiple sequence alignment comprised of sequences representing the standard fungal DNA barcode—ITS [22]. ASAP circumscribes species partitions using an implementation of a hierarchal clustering algorithm based on pairwise genetic distances from single-locus sequence alignments [27]. The pairwise genetic distances are used to build a list of partitions ranked by a score, which is computed using the probabilities of groups to define panmictic species. ASAP provides an objective approach by which to circumscribe relevant species hypotheses as a first step in the process of species delimitation. The multiple sequence alignment was analyzed using the ASAP web server (https://bioinfo.mnhn.fr/abi/public/asap/, accessed on 1 November 2023), with the 'asap-score' considered to select the optimal number of species partitions [27].

Subsequently, ASAP SHs were assessed within a phylogenetic context. New ITS sequences generated for this study were combined with a previously published concatenated five-marker alignment composed of the ITS, nuLSU, mtSSU, RPB1, and RPB2 sequences that represented 382 specimens from the *L. polytropa* complex [15]. ITS sequences were added to the concatenated five-marker alignment using the "–addfragments" function and –multipair alignment strategy in MAFFT v7 [28]. A maximum likelihood (ML) tree was inferred using IQ-TREE [29]. The concatenated alignment was partitioned by loci, with substitution models selected using ModelFinder [30] and nodal support assessed using 2000 ultra-fast bootstrap replicates. SHs inferred using ASAP were compared with the resulting ML topology. We adopted a phylogenetic species criterion, requiring all SHs to be reciprocally monophyletic [31,32]. In cases where ASAP SHs were not recovered as monophyletic in the ML topology, we circumscribed additional monophyletic SHs representing the most inclusive reciprocally monophyletic clades recognized. In three cases distinct SHs that were morphologically similar, originated from the same geographic location, and showed only limited genetic differentiation from their sister clade were combined with the closest related SH.

## 3. Results

For this study, we generated 154 lichen-forming fungal ITS sequences, deposited in GenBank under accession numbers PP104557–PP104710. Of the 154 sequences, 57 represented specimens from Alaska, 86 from specimens collected in Spain, and 11 from species of interest from France. The final ITS alignment—including 154 sequences generated for this study and 356 sequences from an earlier study [15]—comprised 510 sequences and spanned 600 aligned nucleotide position characters (Supplementary File S2).

### 3.1. ASAP Candidate Species Delimitation

The 10 best-scoring ASAP partitions delimited between 85–186 species partitions (SHs, species hypotheses) from the alignment of 510 sequences (Supplementary File S3). The top-ranking species partition scheme comprised 115 species hypotheses (SHs), and we focused on these SHs in subsequent discussions. The ASAP SHs were recovered as reciprocally monophyletic, except for a single ASAP partition that comprised 42 sequences and was highly polyphyletic in the ML topology inferred from a concatenated five-marker data matrix (Figure 2; Supplementary File S4 (alignment)). In three other cases, distinct SHs were morphologically similar and originated from the same geographic location and were combined with the closely related SHs. Ultimately, a total of 123 SHs were circumscribed within the *L. polytropa* complex and are discussed below.

The majority of described taxa within the *L. polytropa* complex that were represented by multiple SHs were recovered as polyphyletic (Figure 2): *L. alpigena* (number [$n$] = 4, in 3 SHs), *L. chlorophaeodes* Nyl. ($n$ = 3, in 2 SHs), *L. dispersoareolata* ($n$ = 5, in 4 SHs), *L intricata* ($n$ = 49, in 17 SHs), *L. polytropa* ($n$ = 351, in 78 SHs), *L. stenotropa* Nyl. ($n$ = 2, in 2 SHs). Four taxa were represented by multiple SHs but specimens representing each taxon were recovered in monophyletic lineages—*L. concolor* Ramond ($n$ = 2, in 2 SHs), *L. solaris* L.S. Yakovchenko and E.A. Davydov ($n$ = 8, in 2 SHs), *L. subcintula* Nyl. ($n$ = 2, in 2 SHs), and *L. subintricata* (Nyl.) Th. Fr. ($n$ = 17, in 3 SHs). Specimens identified as *L. fuscobrunnea* C.W. Dodge and Baker ($n$ = 33), *L. silvae-nigrae* ($n$ = 3) and *L. sommervellii* Paulson ($n$ = 12) were each found to comprise single ASAP SHs.

Newly generated sequences—from specimens collected in Alaska, USA, Spain, and France—were represented in 40 of the 123 SHs, including 32 SHs that had not been represented in earlier studies (Figure 2). Twenty-one of the 123 SHs were found to have intercontinental distributions, e.g., representative specimens occurring on at least two different continents (Table 1), and overall, ca. 45% of sampled specimens belonged to SHs with intercontinental distributions. Based on current sampling of the *L. polytropa* complex, Europe harbored the highest diversity of SHs, followed by North America and Asia (Table 1).

**Table 1.** Summary of distributions patterns—based on current sampling—of all species hypotheses (SHs) in the *L. polytropa* complex. The second column lists the number of SHs that have been documented in each region and the number of sequences ("*n*") representing the SHs. Newly sampled specimens, first reported here, were represented in 40 of the 123 SHs, including 32 SHs that had not been previously sampled.

| | |
|---|---|
| Antarctica | 2 ($n$ = 4) |
| Asia | 21 ($n$ = 91) |
| Europe | 49 ($n$ = 97) |
| North America | 26 ($n$ = 84) |
| South America | 2 ($n$ = 4) |
| Intercontinental | 21 (228) |
| unknown | 2 ($n$ = 2) |
| **Current worldwide sampling** | **123 ($n$ = 510)** |

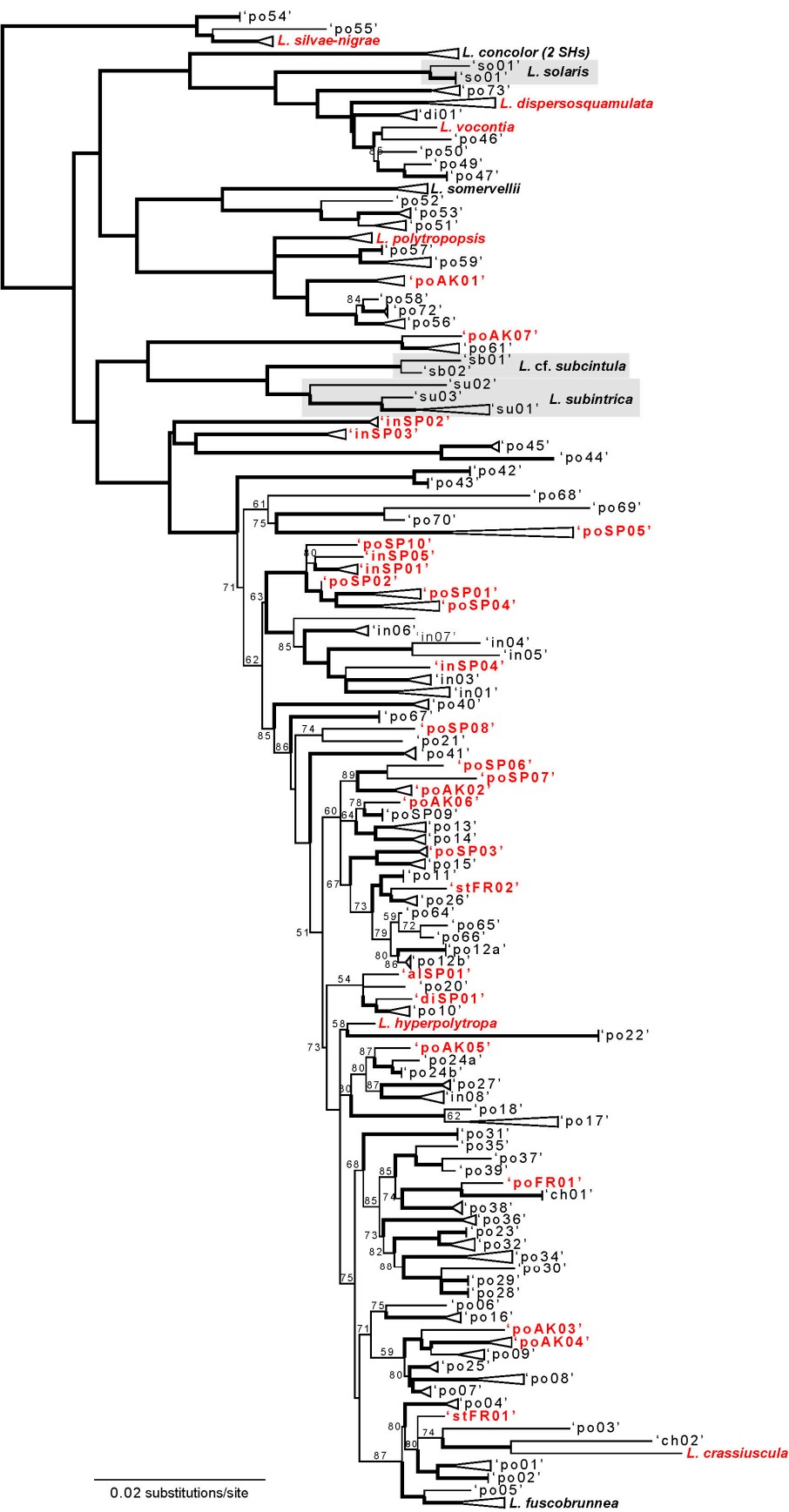

**Figure 2.** Maximum likelihood topology for the *Lecanora polytropa* complex inferred from a concatenated 5-locus data matrix (aligned ITS, nuLSU, mtSSU, RPB1, and RPB2 sequences; Supplementary File S4).

Taxa that were recovered in monophyletic lineages, or represented by only a single specimen, are shown in bold text. All other species hypotheses (SHs) are represented by provisional names: "alXX" represents specimens identified as *L. alpigena*, 'chXX' represents *L. chlorophaeodes*, 'diXX' represents *L. dispersoareolata*, "inXX" represents *L. intricata*, 'poXX' represents *L. polytropa sensu lato*, and "stXX" represents *L. stenotropa*. SHs sampled for the first time in this study are shown in bold, red text. Bootstrap support (BS) values $\geq$ 90% are shown with thickened branches, and BS support between 50–89% is shown at nodes (BS < 50% are not indicated). Tree was rooted with specimens representing the genus *Carbonea* following [15]—outgroup not shown.

### 3.1.1. ASAP Candidate Species at Two Sites in the Pacific Coast Ranges

Specimens collected from the Chugach and the Kenai Mountains in Alaska (Figure 1A) were recovered in a total of 17 SHs. The 24 sequences from specimens collected from the site in the Chugach Mountains were recovered in 11 SHs, while the 33 sequences from specimens collected at the site in the Kenai Mountains were recovered in 9 SHs (Table 2). The majority of SHs represented in the Alaskan samples—11 of 17—were also found in populations outside of North America, including 7 SHs that also occurred in newly sampled populations in Spain (Supplementary File S1). Despite the proximity of the sites in the Chugach and Kenai Mountains, only 3 of the 17 SHs were shared between the 2 sampled areas. Of the 17 SHs comprising specimens from Alaska, 6 were unique to the sampled sites and not found in other current worldwide sampling.

**Table 2.** Summary of the geographic distributions of the species hypotheses (SHs) sampled at two sites in the Pacific Coast Ranges in southern Alaska, USA.

| Distribution of SHs | Chugach Mountains | Kenai Mountains |
|---|---|---|
| Intercontinental | 8 [1] | 6 [1] |
| Unique to site | 3 | 3 |
| Total SHs | 11 | 9 |

[1] Three species hypotheses (SHs) with intercontinental distributions were shared between sampled locations in the Chugach and Kenai Mountains, and 11 of the 17 SHs sampled from the Pacific Coast Ranges were found to have Intercontinental distributions.

### 3.1.2. ASAP Candidate Species at Three Sites in Montane Habitats in Spain

Specimens collected from the Central and Iberian Systems, and Cantabrian Mountains (Figure 1B) were recovered in a total of 30 SHs (Table 3). The 33 sequences from specimens collected from the Sierra de Guadarrama (Central System) were recovered in 18 SHs, the 37 sequences from specimens collected from Laguna Negra (Iberian System) were recovered in 14 SHs, and 15 sequences from specimens collected from Ponga Biosphere Reserve (Cantabrian Mountains) were recovered in 10 SHs. Each sampling area included SHs with intercontinental and European distributions, in addition to a significant proportion of SHs unique to the area (Table 3).

**Table 3.** Summary of the geographic distributions of the species hypotheses (SHs) sampled at three sites in Mediterranean and Atlantic montane habitats in Spain.

| Distribution of SHs | Sierra de Guadarrama | Laguna Negra y Circos Glaciares de Urbion | Ponga Biosphere Reserve |
|---|---|---|---|
| Intercontinental SHs | 6 | 6 | 2 |
| SHs unique to site | 10 | 6 | 4 |
| Regional SHs | 2 | 2 | 4 |
| Total SHs | 18 | 14 | 10 |

### 3.1.3. ASAP Candidate Species in Species of Interest from France

The 11 sequences generated from specimens of taxonomic interest collected in France were recovered in nine ASAP SHs. *Lecanora crassiuscula* (*n* = 1), *L. hyperpolytropa* (*n* = 1), and *L. vocontia* (*n* = 1) were each recovered as distinct, single ASAP SHs (Supplementary

File S3). Sequences from French specimens representing *L. dispersoareolata* (*n* = 1) and *L. dispersosquamulata* ad int. (*n* = 2) were recovered together in a single ASAP SH. The single sequence representing *L. hyperpolytropa* was initially recovered in a single SH, along with 41 other sequences from specimens originally identified as *L. polytropa s. lat.* However, this was the single SH that was not recovered as monophyletic and subsequently split into multiple SHs based on a phylogenetic species delimitation criterion, e.g., requiring reciprocal monophyly of all SHs. The single sequence representing *L. polytropopsis* was recovered in an SH along with six other sequences from specimens originally identified as *L. polytropa s. lat.* Subsequent morphological determinations of these specimens fit with the description of *L. polytropopsis*, and this SH was hereafter considered to represent *L. polytropopsis*, with representatives from France, Spain, and Switzerland (Supplementary File S1). The single French specimen determined as *L. polytropa* belonged to a single SH (Supplementary File S3). The two specimens from France determined as *L. stenotropa* were recovered in two separate single SHs morpho-anatomically distinguished as "morpho. stenotropa" (MBe 7195) and "morpho. large apothecia" (MBe6353) (Supplementary File S3). This separation of the two *L. stenotropa* forms must be confirmed with additional specimen sampling and additional DNA sequence data.

## 4. Discussion

In this study, we focused on generating ITS sequence data—the standard barcoding marker for fungi—from populations of members of the *Lecanora polytropa* complex *sensu* [15] in the Pacific Coast Ranges in Alaska, USA, and three Mediterranean and Atlantic montane habitats in Spain. Our investigation unveiled high diversity within both regions, including previously unsampled candidate species (SHs). We also observed only limited overlap in SHs at all sampled scales, e.g., among individual sampling sites and between geographic regions (Tables 1–3). Some taxonomic novelties have recently been proposed for members of this complex [16], and, here, we generated sequence data for the first time for some of these new species. The sequence data generally support the evolutionary independence of the recent taxonomic proposals (Figure 2), supporting the perspective that an integrative taxonomy may ultimately help resolve the potential taxonomic conundrums within this extremely diverse lineage. An incredible range of phenotypic variation was observed in our newly collected specimens from Pacific Coast Ranges in Alaska, USA, and montane habitats in Spain (Figure 3). While a detailed taxonomic investigation was beyond the scope of this study, our results highlight the need for thorough specimen sampling at even local scales. In many cases, multiple distinct lineages representing members of the *L. polytropa* complex may occur side by side on a single specimen. While this demands careful attention to document the targeted thalli, these mixed collections also provide potentially important opportunities to compare phenotypes of different SHs growing in very similar conditions and to draw taxonomic conclusions.

Species circumscription and specimen identification in members of the *L. polytropa* complex has been challenging [16,19,20], although recent taxonomic work has provided critical insights into diagnostic phenotypic traits [16]. The distinction between *L. polytropa sensu stricto* (*s. str.*) and *L. polytropa sensu lato* (*s. lat.*) remains unclear, as specimens identified as *L. polytropa* are recovered as highly polyphyletic [15]. To help resolve the issue of which lineage represents *L. polytropa s. str.*, sampling and sequencing of specimens from the region where the type specimen originated—near Göttingen, Harz, Germany—will be essential to differentiate *L. polytropa s. str.* from the broader group of morphologically and anatomically similar but potentially distinct species-level lineages. Alternatively, advances in sequencing technologies have facilitated the sequencing of old herbarium specimens, and it may be possible to successfully generate sequence data from the type specimen [33].

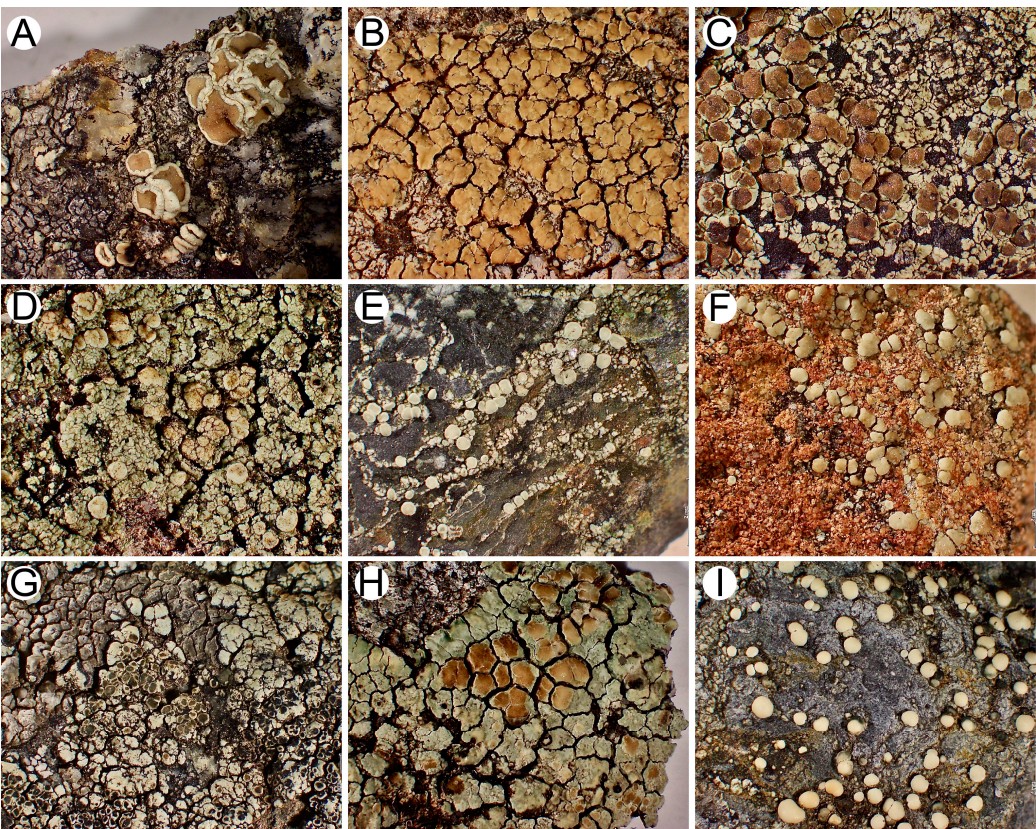

**Figure 3.** Examples of the range of sampled morphological diversity in the specimens collected from the Pacific Coast Ranges in Alaska, USA, and montane habitats in Spain. (**A**) *Lecanora sylvae-nigrae* (*Pérez-Ortega 11817*), (**B**) *L. polytropa s. lat.* 'poSP01' (*Pérez-Ortega 11601*), (**C**). *L. polytropa s. lat.* 'poSP03' (*Pérez-Ortega 11807*), (**D**) *L. polytropopsis* (*Pérez-Ortega 11571*), (**E**) *L. polytropa s. lat.* 'poAK02' (*Leavitt 22126*), (**F**) *L. polytropa s. lat.* 'po09' (*Leavitt 22108*), (**G**) *intricata s. lat.* 'in01' (*Leavitt 22114*), (**H**) *L. intricata s. lat.* 'inSP02' (*Pérez-Ortega 11550*), (**I**) *L. polytropa s. lat.* 'po08' (*Leavitt 22121*). All specimens are housed at the Herbarium of Non-Vascular Cryptogam, Brigham Young University, Provo, Utah, USA (BRY).

Based on current sampling, Europe harbors the highest diversity of members of the *L. polytropa* complex (Table 1). However, the observed richness in Europe may partly stem from variations in research efforts in different geographical regions. Thus, it may not entirely reflect actual species richness. These observations prompt further investigation to determine whether the observed patterns are solely attributable to sampling intensity or are influenced by other biogeographical factors.

Like most regions, the French Alps harbor poorly understood diversity within the *L. polytropa* complex. Recent work in this region provides critical insight into how the taxonomy of this extremely diverse lineage may, in part, be resolved [16]. Notably, four taxa, *L. dispersoareolata*, *L. hyperpolytropa*, *L. polytropopsis*, and *L. stenotropa*, are common in the Alps, each with distinct combinations of morphological and anatomical characteristics. Our results support the recognition of two commonly occurring species, *L. hyperpolytropa* and *L. polytropopsis*. These species can be readily distinguished from *L. polytropa s. lat.* [16]. However, some specimens representing *L. dispersoareolata* and *L. dispersosquamulata* ad int., initially distinguished with uncertainty in previous research, were grouped here into a single SH. *Lecanora dispersoareolata* appears to encompass various chemotypes and morphotypes that warrant further study. The separation of the two specimens representing *L. stenotropa*, based on apothecia and spore dimensions, were recovered in two distinct ASAP SHs. For *L. dispersoareolata*, *L. dispersosquamulata* ad int., and *L. stenotropa*, further

analyses, incorporating a larger sample size and additional genetic data, will be necessary to definitively resolve their taxonomic status.

Our work expands the known distribution of *L. polytropopsis* beyond the subalpine and alpine zones in France, including the Alps and the Pyrenees. Here, *L. polytropopsis* was recovered in two additional mountain ranges in Spain—the Iberian system and the Cantabrian Mountains. *Lecanora crassiuscula* appears to be strictly Mediterranean, but its distribution remains incompletely understood, possibly encompassing the circum-Mediterranean region. Here this taxon was recovered as a distinct ASAP SH, supporting its delimitation as a distinct species within the *L. polytropa* complex. What we refer to as *L. polytropa s. lat.* appears to have broader, more ambiguous distributions encompassing a wide range of altitudes, spanning inland to coastal habitats to inland montane regions. This variation aligns with the findings of Roux et al. [16]. Future work considering the ecological context and habitat preference for distinct SHs representing *L. polytropa s. lat.* may help better understand the circumscription of species-level SHs in this group.

Our sampling of members of the *L. polytropa* complex in the Pacific Coast Ranges, Alaska, USA, and montane habitats in Spain provides novel insights into how genetic diversity is structured in this species complex. Strikingly, all sampled sites displayed a high diversity of SHs, ranging from 9 to 18 SHs per site. In some cases, multiple SHs occurred on the same rock surface within mm to cm of each other. Furthermore, each site harbored a significant proportion of unique SHs that were not shared among other sites, many of which had not been represented in previous sampling efforts [15]. Six of the seventeen SHs sampled in Alaska and sixteen of the twenty-six SHs sampled in Spain, plus one SH shared between Alaska and Spain, had not been sampled in earlier studies (Supplementary File S1).

Similar to earlier work [15], here we documented a mix of local local/endemic SHs and a limited number of cosmopolitan lineages (Tables 1 and 2). Of the 123 total ASAP SHs, 21 have documented intercontinental distributions, and our sampling efforts in Alaska and Spain collectively captured 12 of the known 21 SHs with cosmopolitan distributions (Supplementary File S1). We noted a higher proportion of intercontinental SHs in the two sites sampled in the Pacific Coast Ranges, Alaska, USA, relative to those with intercontinental distributions in Spain (Tables 2 and 3). This finding raises intriguing questions about potential factors influencing distributions. Environmental conditions in Alaska, such as a more recent glaciation history [34], and relatively recently exposed suitable habitat may result in a higher probability of recent colonization of cosmopolitan SHs, and we speculate that cosmopolitan lineages may have unknown dispersal adaptations. However, we have also observed high SH diversity and some unique SHs that have not yet been sampled beyond the Pacific Coast Ranges (Table 2), and these results suggest that glacial refugia may also play an important role in the diversity of members of the *L. polytropa* complex in the Alaskan Pacific Coast Ranges [35].

Our sampling of three mountain ranges in Spain captured a total of twenty-six SHs, with a significant number of SHs unique to each site (Table 3). The Cantabrian Mountains belong to the Atlantic ecoregion, while the Sierra de Guadarrama, which is part of the Central System, and the Iberian System, in the north of which the Laguna Negra Natural Park is located, belong to the Mediterranean ecoregion [36]. The degree of historical connection between these mountain ranges seems to vary depending on the organisms, with the Central and Iberian systems showing some degree of connection, though they are isolated from the Cantabrian Range for some organisms [37], and the northern part of the Iberian System being an ecotone for other taxa bridging the Atlantic and Mediterranean regions [38]. The influence of different ecoregions in determining the occurrence of various members of the *L. polytropa* complex remains unexplored. Furthermore, a broader contextual analysis is needed in order to explore the role of the Iberian Peninsula as a glacial refugium in past periods for lichen-forming fungi, including members of the *L. polytropa* complex [39,40].

The documented intercontinental distribution of at least 21 SHs (Table 1) suggests the potential for broad dispersal capacity for members of the *L. polytropa* complex. We speculate that perhaps some lichen-forming fungal lineages in the *L. polytropa* complex may

have specialized photobionts and that photobiont availability may limit the distribution of some lineages within this group. Interactions with distinct photobionts may vary even among closely related lichen-forming fungal species, ultimately influencing lichen distributions [41–43]. For example, the closely related species *Letharia lupina* and *Letharia vulpina* (Parmeliaceae) form mutually exclusive partnerships with distinct *Trebouxia* photobiont lineages [44]. Future research will be essential to an assessment of the potential role of symbiont availability and interactions when influencings the geographic and ecological distributions of different lineages within the *L. polytropa* complex.

With the limited number of shared SHs among mountain ranges, both in those sampled in Spain (Table 3) and in the Pacific Coast Ranges (Table 2), it is challenging to draw definitive conclusions related to the role of glacial refugia. The nuances observed here emphasize the need for a more comprehensive investigation into the biogeographic origins of the observed diversity.

**Supplementary Materials:** The following supporting information can be downloaded at: https://www.mdpi.com/article/10.3390/d16020088/s1, Supplementary File S1: List of specimens from the "*L. polytropa* complex" included in this study, data include voucher number and herbarium for specimens collected for this study and GenBank accession numbers for sequences from GenBank; Supplementary File S2: Multiple sequence alignment (MSA) of the internal transcribed spacer region (ITS; comprised of 510 aligned sequences) used in the ASAP species delimitation analysis. Supplementary File S3: ASAP scores and rankings for the '*L. polytropa* complex' inferred from a multiple sequence alignment of the internal transcribed spacer region (ITS, standard fungal barcode marker) and comprising 510 sequences. The ten best partitions and coinciding ASAP scores and rankings (panel on left) and graphical representation of all ASAP scores (center and right panels), highlighting the ten best partitions shown in the panel. Supplementary File S4: Concatenated five-marker (ITS, nuLSU, RPB1, RPB2, mtSSU) alignment representing the *L. polytropa group* and the putative sister clade *Carbonea*. The five-marker dataset comprised 536 specimens and spanned 3913 aligned nucleotide position characters—ITS ($n = 533$; 626 bp MSA), nuLSU ($n = 122$; 843 bp MSA (ambiguous sites removed)), RPB1 ($n = 117$; 816 bp MSA), RPB2 ($n = 105$; 828 bp MSA), and mtSSU ($n = 112$; 801 bp MSA (ambiguous sites removed)).

**Author Contributions:** Conceptualization, S.D.L. and J.W.; methodology, S.D.L. and J.W.; formal analysis, J.W.; investigation, S.D.L., S.P.-O., C.R. and M.B.; resources, S.D.L., S.P.-O., C.R. and M.B.; data curation, S.D.L. and J.W.; writing—original draft preparation, S.D.L. and J.W.; writing—review and editing, S.D.L., S.P.-O., C.R. and M.B.; visualization, S.D.L. and J.W.; supervision, S.D.L.; project administration, S.D.L.; funding acquisition, S.D.L., S.P.-O., C.R. and M.B. All authors have read and agreed to the published version of the manuscript.

**Funding:** This research was funded by the Museum of Life Sciences at Brigham Young University, Provo, UT, USA.

**Institutional Review Board Statement:** Not applicable.

**Data Availability Statement:** Newly generated ITS sequences are deposited in GenBank under accession numbers PP104557-PP104710, and multiple sequence alignments are provided as Supplementary Files S2 and S4. Multiple sequence alignments used in this study are provided as Supplementary Files.

**Acknowledgments:** We thank four anonymous reviewers whose comments helped improve the study. Jeffrey Clancy provided invaluable conceptual discussion and feedback, and Pierce Adams' help with specimen photography is greatly appreciated. We also thank the Escalante River Watershed Partnership for facilitating a portion of the fieldwork.

**Conflicts of Interest:** The authors declare no conflict of interest. The funders had no role in the design of the study; in the collection, analyses, or interpretation of data; in the writing of the manuscript; or in the decision to publish the results.

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
