# Peer review of "Diverse Communities of Endemic and Cosmopolitan Lineages at Local Sites in the Lecanora polytropa Aggregate (Ascomycota)"

_diversity, doi:10.3390/d16020088_

Round 1
Reviewer 1 Report
Comments and Suggestions for Authors
This well-written manuscript is based on scientifically sound hypotheses, methods and hypothesis testing.
The results presented in this manuscript are based on a very large data set (510 ITS sequences, 600 bp long alignment) from several parts of the geographical distribution of the Lecanora polytropa complex, which is an important strength. I personally think it is enough to encompass the genetic variability using only one fungal genetic marker (ITS). However, to avoid readers from dismissing the results (arguing that “more genes, more samples” will give better results), it is necessary to better justify the use of this marker somewhere in the text. Also, it would be interesting with some dicussion of mycobiont genetics compared to photobiont diversity, and how that might influence the resulting phenotypic expression (the lichen).
Specific comments
Line 122: a reference is lacking “(CITE)”
Line 263: “each with distict characteristics”, are the distinct characteristics anatomical, morphological, chemical…?
Line 265: “diagnosed” changed to distinguished
Line 278: “previously unassumed” Is unassumed really the correct terminology?
Line 322: “difference” change to different (I think)
Line 357 – 441: The references need to be checked in detail, I find several small errors
Line 363: One example, Nash, T.H. III
Line 394: Several typographical errors in the reference
Line 412: actually is not in press anymore. Fungal Diversity 2016, 78, 293-304.
Author Response
Reviewer 1
This well-written manuscript is based on scientifically sound hypotheses, methods and hypothesis testing.
The results presented in this manuscript are based on a very large data set (510 ITS sequences, 600 bp long alignment) from several parts of the geographical distribution of the Lecanora polytropa complex, which is an important strength. I personally think it is enough to encompass the genetic variability using only one fungal genetic marker (ITS). However, to avoid readers from dismissing the results (arguing that “more genes, more samples” will give better results), it is necessary to better justify the use of this marker somewhere in the text. Also, it would be interesting with some dicussion of mycobiont genetics compared to photobiont diversity, and how that might influence the resulting phenotypic expression (the lichen).
Based on reviewers’ comments, we have now added an analysis of a multi-locus sequence data matrix to better place our results within a phylogenetic context. Based on the phylogenetic analysis, one (of 115) species hypotheses inferred from the single locus data matrix was further subdivided into multiple species hypotheses. The results and discussion have been revised throughout to reflect these changes.
New text in Methods:
“Subsequently, ASAP SHs were assessed within a phylogenetic context. New ITS sequences generated for this study were combined with a previously published concatenated five-marker alignment comprised of the ITS, nuLSU, mtSSU, RPB1, and RPB2 sequences that represented 382 specimens from the L. polytropa complex [15]. ITS sequences were added to the concatenated five-marker alignment using the “--addfragments” function and –multipair alignment strategy in MAFFT v7 [28]. A maximum likelihood (ML) tree was inferred from using IQ-TREE [29]. The concatenated alignment was partitioned by loci, with substitution models selected using ModelFinder [30] and nodal support assessed using 2000 ultra-fast bootstrap. SHs inferred using ASAP were compared to the resulting ML topology. We adopted a phylogenetic species criterion, requiring all SHs to be reciprocally monophyletic [31,32]. In cases, where ASAP SHs were not recovered as monophyletic in the ML topology, we circumscribed additional monophyletic SHs representing the most inclusive reciprocally monophyletic clades recognized. In three cases distinct SHs that were morphologically similar, originated from the same geographic location, and show only limited genetic differentiation from its sister clade were combined with the closest related SH.”
New and revised text in Results:
“The ten best-scoring ASAP partitions delimited between 85–186 species partitions (SHs, species hypotheses) from the alignment of 510 sequences (Supplementary file S3). The top-ranking species partition scheme comprised 115 species hypotheses (SHs), and we focused on these SHs in subsequent discussions. The ASAP SHs were recovered as reciprocally monophyletic, except for a single ASAP partition that comprised 42 sequences and was highly polyphyletic in the ML topology inferred from a concatenated five-marker data matrix (Fig. 2; supplementary file S3 [alignment]). In three other cases, distinct SHs were morphologically similar and originated from the same geographic location were combined the closely related SHs. Ultimately, a total of 123 SHs were circumscribed within the L. polytropa complex and discussed below.
The majority of described taxa within the L. polytropa complex that were represented by multiple SHs were recovered as polyphyletic (Fig 2): L. alpigena (number [n]=4, in 3 SHs), L. chlorophaeodes Nyl. (n=3, in 2 SHs), L. dispersoareolata (n=5, in 4 SHs), L intricata (n=49, in 17 SHs), L. polytropa (n=351, in 78 SHs), L. stenotropa Nyl. (n=2, in 2 SHs). Four taxa were represented by multiple SHs but specimens representing each taxon were recovered in monophyletic lineages – L. concolor Ramond (n=2, in 2 SHs), L. solaris L.S. Yakovchenko & E.A. Davydov (n=8, in 2 SHs), L. subcintula Nyl. (n=2, in 2 SHs), and L. subintricata (Nyl.) Th. Fr. (n=17, in 3 SHs). Specimens identified as L. fuscobrunnea C.W. Dodge & Baker (n=33), L. silvae-nigrae (n=3) and L. sommervellii Paulson (n=12) were each, respectively, found to comprise single ASAP SHs.”
New figure and caption: “Figure 2. Maximum likelihood topology for the Lecanora polytropa complex inferred from a concatenated 5-locus data matrix (aligned ITS, nuLSU, mtSSU, RPB1, and RPB2 sequences; supplementary file S3). Taxa that were recovered in monophyletic lineages, or represented by only a single specimen, are shown in bold text. All other species hypotheses (SHs) are represented by provisional names: “alXX” represent specimens identified as L. alpigena, ‘chXX’ representing L. chlorophaeodes, ‘diXX’ representing L. dispersoareolata, “inXX” representing L. intricata, and ‘po’ representing L. polytropa sensu lato. SHs sampled for the first time in this study are shown in bold, red text. Bootstrap support (BS) values ≥ 90% are shown with thickened branches, and BS support between 50–89% is shown at nodes (BS < 50% are not indicated). Tree was rooted with specimens representing the genus Carbonea following [15] – outgroup not shown.
Specific comments
Line 122: a reference is lacking “(CITE)”
added
Line 263: “each with distict characteristics”, are the distinct characteristics anatomical, morphological, chemical…?
Revised text: “… with distinct combinations of morphological and anatomical characteristics.”
Line 265: “diagnosed” changed to distinguished
Changed to “distinguished”
Line 278: “previously unassumed” Is unassumed really the correct terminology?
Revised text: “resulting in an unexpected cosmopolitan distribution for this taxon.”
Line 322: “difference” change to different (I think)
Corrected
Line 357 – 441: The references need to be checked in detail, I find several small errors
We have carefully can through all the references and made appropriate corrections throughout.
Line 363: One example, Nash, T.H. III
Corrected
Line 394: Several typographical errors in the reference
Corrected
Line 412: actually is not in press anymore. Fungal Diversity 2016, 78, 293-304.
Corrected
Reviewer 2 Report
Comments and Suggestions for Authors
Dear Editor and Authors,
The study offers new and original data on the diversity of lichenized fungi belonging to Lecanora polytropa species complex, which is a taxonomically problematic group. The manuscript deals with the diversity of the members at local scales (samples from the southern Alaska and Spain - two regions that have not been investigated in earlier studies) and compared with the diversity data from the previous studies. The manuscript brings first genetic sequence data on some recently described species, which had been missing so far. The Authors also highlighted several open matters for future research.
The results are clearly presented and appropriately discussed. The text is well-structured. The methodology has been adequately described. The figures and tables are mostly clear and properly show the data. The references cited are relevant and refer to mostly recent publications, without an excessive number of self-citations. The manuscript is overall interesting and valuable for publication. In the following section there are few minor points which the Authors can consider.
Line 122: missing reference: „...ITS1f with ITS4 (CITE). The...“
Line 123: „follow“ to be corrected as „follows“
Line 155,156: missing verb in the sentence
Line 159: The meaning of abbreviation „n“ is not described. Since this abbreviation is repeatedly used, I suggest to clarify it also in the legends of all tables.
Line 229: missing a dot at the end of the paragraph
Line 322: „...difference ecoregions...“ to be corrected as „different ecoregions“
Author Response
Reviewer 2
The study offers new and original data on the diversity of lichenized fungi belonging to Lecanora polytropa species complex, which is a taxonomically problematic group. The manuscript deals with the diversity of the members at local scales (samples from the southern Alaska and Spain - two regions that have not been investigated in earlier studies) and compared with the diversity data from the previous studies. The manuscript brings first genetic sequence data on some recently described species, which had been missing so far. The Authors also highlighted several open matters for future research.
The results are clearly presented and appropriately discussed. The text is well-structured. The methodology has been adequately described. The figures and tables are mostly clear and properly show the data. The references cited are relevant and refer to mostly recent publications, without an excessive number of self-citations. The manuscript is overall interesting and valuable for publication. In the following section there are few minor points which the Authors can consider.
Line 122: missing reference: „...ITS1f with ITS4 (CITE). The...“
added
Line 123: „follow“ to be corrected as „follows“
corrected
Line 155,156: missing verb in the sentence
The verb in this sentence is “delimited”
Line 159: The meaning of abbreviation „n“ is not described. Since this abbreviation is repeatedly used, I suggest to clarify it also in the legends of all tables.
Revised to “(number [n]=…” and in Table 1 we added “The second column lists the number of SHs that have been documented in each region and the number of sequences (“n”) representing the SHs.”
Line 229: missing a dot at the end of the paragraph
corrected
Line 322: „...difference ecoregions...“ to be corrected as „different ecoregions“
corrected
Reviewer 3 Report
Comments and Suggestions for Authors
Dear authors,
1. Current study is a nice piece of work and I enjoyed reading the manuscript. However, the manuscript required major revisions to improve its quality.
2. Manuscript aimed to investigate diversity across multiple spatial scales of lichen-forming Lecanora polytropa complex, but diversity related biological and ecological events are not clearly discussed.
3. Line 25, 26- “Our sequence data generally supported the recent taxonomic proposals, and an integrative taxonomy may help resolve the taxonomic conundrums within this hyper-diverse lineage”
Line 236, 237- “Some taxonomic novelties have recently been proposed for members of this complex [16], and here, we generated sequence data for the first time for some of these new species”
To support the above statements, related phylogenetic analyses are required to provide within the manuscript.
4. Phenotypic variations are major component in diversity. Therefore, manuscript need to improve adding some major morphological and physiological character descriptions on taxa collected from different regions and compare them with previous literature records.
Author Response
Reviewer 3
- Current study is a nice piece of work and I enjoyed reading the manuscript. However, the manuscript required major revisions to improve its quality.
We appreciate the thoughtful, constructive feedback. Thank you for your comments – these have been very helpful for improving our manuscript.
- Manuscript aimed to investigate diversity across multiple spatial scales of lichen-forming Lecanora polytropacomplex, but diversity related biological and ecological events are not clearly discussed.
We recognize the limitation highlighted here and agree that the connections between genetic diversity and biological/ecological events are not well characterized in this study. Given the relatively high number of candidate species occurring even at the smallest spatial scale, coupled with the sparse geographic sampling, our ability to infer the role of biogeographical events is severely limited. The best we could do was highlight the high diversity within and among sites.
- Line 25, 26- “Our sequence data generally supported the recent taxonomic proposals, and an integrative taxonomy may help resolve the taxonomic conundrums within this hyper-diverse lineage”
Line 236, 237- “Some taxonomic novelties have recently been proposed for members of this complex [16], and here, we generated sequence data for the first time for some of these new species”
To support the above statements, related phylogenetic analyses are required to provide within the manuscript.
Based on reviewers’ comments, we have now added an analysis of a multi-locus sequence data matrix to better place our results within a phylogenetic context. Based on the phylogenetic analysis, one (of 115) species hypotheses inferred from the single locus data matrix was further subdivided into multiple species hypotheses. The results and discussion have been revised throughout to reflect these changes.
New text in Methods:
“Subsequently, ASAP SHs were assessed within a phylogenetic context. New ITS sequences generated for this study were combined with a previously published concatenated five-marker alignment comprised of the ITS, nuLSU, mtSSU, RPB1, and RPB2 sequences that represented 382 specimens from the L. polytropa complex [15]. ITS sequences were added to the concatenated five-marker alignment using the “--addfragments” function and –multipair alignment strategy in MAFFT v7 [28]. A maximum likelihood (ML) tree was inferred from using IQ-TREE [29]. The concatenated alignment was partitioned by loci, with substitution models selected using ModelFinder [30] and nodal support assessed using 2000 ultra-fast bootstrap. SHs inferred using ASAP were compared to the resulting ML topology. We adopted a phylogenetic species criterion, requiring all SHs to be reciprocally monophyletic [31,32]. In cases, where ASAP SHs were not recovered as monophyletic in the ML topology, we circumscribed additional monophyletic SHs representing the most inclusive reciprocally monophyletic clades recognized. In three cases distinct SHs that were morphologically similar, originated from the same geographic location, and show only limited genetic differentiation from its sister clade were combined with the closest related SH.”
New and revised text in Results:
“The ten best-scoring ASAP partitions delimited between 85–186 species partitions (SHs, species hypotheses) from the alignment of 510 sequences (Supplementary file S3). The top-ranking species partition scheme comprised 115 species hypotheses (SHs), and we focused on these SHs in subsequent discussions. The ASAP SHs were recovered as reciprocally monophyletic, except for a single ASAP partition that comprised 42 sequences and was highly polyphyletic in the ML topology inferred from a concatenated five-marker data matrix (Fig. 2; supplementary file S3 [alignment]). In three other cases, distinct SHs were morphologically similar and originated from the same geographic location were combined the closely related SHs. Ultimately, a total of 123 SHs were circumscribed within the L. polytropa complex and discussed below.
The majority of described taxa within the L. polytropa complex that were represented by multiple SHs were recovered as polyphyletic (Fig 2): L. alpigena (number [n]=4, in 3 SHs), L. chlorophaeodes Nyl. (n=3, in 2 SHs), L. dispersoareolata (n=5, in 4 SHs), L intricata (n=49, in 17 SHs), L. polytropa (n=351, in 78 SHs), L. stenotropa Nyl. (n=2, in 2 SHs). Four taxa were represented by multiple SHs but specimens representing each taxon were recovered in monophyletic lineages – L. concolor Ramond (n=2, in 2 SHs), L. solaris L.S. Yakovchenko & E.A. Davydov (n=8, in 2 SHs), L. subcintula Nyl. (n=2, in 2 SHs), and L. subintricata (Nyl.) Th. Fr. (n=17, in 3 SHs). Specimens identified as L. fuscobrunnea C.W. Dodge & Baker (n=33), L. silvae-nigrae (n=3) and L. sommervellii Paulson (n=12) were each, respectively, found to comprise single ASAP SHs.”
New figure and caption: “Figure 2. Maximum likelihood topology for the Lecanora polytropa complex inferred from a concatenated 5-locus data matrix (aligned ITS, nuLSU, mtSSU, RPB1, and RPB2 sequences; supplementary file S3). Taxa that were recovered in monophyletic lineages, or represented by only a single specimen, are shown in bold text. All other species hypotheses (SHs) are represented by provisional names: “alXX” represent specimens identified as L. alpigena, ‘chXX’ representing L. chlorophaeodes, ‘diXX’ representing L. dispersoareolata, “inXX” representing L. intricata, and ‘po’ representing L. polytropa sensu lato. SHs sampled for the first time in this study are shown in bold, red text. Bootstrap support (BS) values ≥ 90% are shown with thickened branches, and BS support between 50–89% is shown at nodes (BS < 50% are not indicated). Tree was rooted with specimens representing the genus Carbonea following [15] – outgroup not shown.
- Phenotypic variations are major component in diversity. Therefore, manuscript need to improve adding some major morphological and physiological character descriptions on taxa collected from different regions and compare them with previous literature records.
New text and figure:
“An incredible range of phenotypic variation was observed in our newly collected specimens from Pacific Coast Ranges in Alaska, USA, and montane habitats in Spain (Fig. 3). While a detailed taxonomic investigation was beyond the scope of this study, our results highlight the need for thorough specimen sampling at even local scales. In many cases, multiple distinct lineages representing members of the L. polytropa complex may occur side by side on a single specimen. While this demands careful attention to document the targeted thalli, these mixed collections also provide potentially important opportunities to compare phenotypes of different SHs growing in very similar conditions and draw taxonomic conclusions.
Figure 3. Examples of the ranged of sampled morphological diversity in the specimens collected from the Pacific Coast Ranges in Alaska, USA, and montane habitats in Spain. A. Lecanora sylvae-nigrae (Pérez-Ortega 11817); B. L. polytropa s. lat. ‘poSP01’ (Pérez-Ortega 11601); C. L. polytropa s. lat. ‘poSP03’ (Pérez-Ortega 11807); 11807; D. L. polytropopsis (Pérez-Ortega 11571); E. L. polytropa s. lat. ‘poAK02’ (Leavitt 22126); F. L. polytropa s. lat. ‘po09’ (Leavitt 22108); G. intricata s. lat. ‘in01’ (Leavitt 22114); H. L. intricata s. lat. ‘inSP02’ (Pérez-Ortega 11550); I. L. polytropa s. lat. ‘po08’ (Leavitt 22121). All specimens are housed at the Herbarium of Non-Vascular Cryptogam, Brigham Young University, Provo, Utah, USA (BRY).
Reviewer 4 Report
Comments and Suggestions for Authors
Comments to the manuscript “Diverse communities of endemic and cosmopolitan lineages at local sites in the Lecanora polytropa aggregate (Ascomycota)” by Woodhouse et al.
In the submitted manuscript, the diversity of species within the lichen-forming fungal Lecanora polytropa complex is analyzed. The analysis carried out included specimens from the Pacific Coast Ranges in southern Alaska (USA), montane habitats in Spain and from several localities in France. The ITS region (ITS1-5.8S-ITS2) was used as standard barcoding in fungi. With the ITS sequences of 510 specimens, a multiple alignment was generated which was used to infer species candidates using the sequence-based species delimitation approach (ASAP algorithm). The results indicate 115 candidate species (SHs) within the L. polytropa complex. Of these, 29 were sampled for the first time and 21 have intercontinental distribution. In addition, 36 SHs with high endemism were found from Alaska and Spain. The authors conclude that their “…sequence data generally supported the recent taxonomic proposals, and an integrative taxonomy may help resolve the taxonomic conundrums within this hyper-diverse lineage.”
The submitted manuscript is well written, the methodology is properly described, the results are clearly explained, and the discussion adequately addressed. The only comment is not forgotten to properly add the GenBank accession numbers of the ITS sequences generated in the final version of the manuscript.
Thus, I consider that the document is suitable for its publication in the Diversity Journal.
Author Response
Reviewer 4
In the submitted manuscript, the diversity of species within the lichen-forming fungal Lecanora polytropa complex is analyzed. The analysis carried out included specimens from the Pacific Coast Ranges in southern Alaska (USA), montane habitats in Spain and from several localities in France. The ITS region (ITS1-5.8S-ITS2) was used as standard barcoding in fungi. With the ITS sequences of 510 specimens, a multiple alignment was generated which was used to infer species candidates using the sequence-based species delimitation approach (ASAP algorithm). The results indicate 115 candidate species (SHs) within the L. polytropa complex. Of these, 29 were sampled for the first time and 21 have intercontinental distribution. In addition, 36 SHs with high endemism were found from Alaska and Spain. The authors conclude that their “…sequence data generally supported the recent taxonomic proposals, and an integrative taxonomy may help resolve the taxonomic conundrums within this hyper-diverse lineage.”
The submitted manuscript is well written, the methodology is properly described, the results are clearly explained, and the discussion adequately addressed. The only comment is not forgotten to properly add the GenBank accession numbers of the ITS sequences generated in the final version of the manuscript.
Thus, I consider that the document is suitable for its publication in the Diversity Journal.
We appreciate the careful reading of our manuscript and positive feedback. We have added the GenBank Accession numbers of the newly generated sequences.